# Regulation of Tomato Specialised Metabolism after Establishment of Symbiosis with the Endophytic Fungus *Serendipita indica*

**DOI:** 10.3390/microorganisms10010194

**Published:** 2022-01-16

**Authors:** Fani Ntana, Sean R. Johnson, Björn Hamberger, Birgit Jensen, Hans J. L. Jørgensen, David B. Collinge

**Affiliations:** 1Department of Plant and Environmental Sciences and Copenhagen Plant Science Centre, University of Copenhagen, Thorvaldsensvej 40, 1871 Copenhagen, Denmark; fntana@envs.au.dk (F.N.); bje@plen.ku.dk (B.J.); hjo@plen.ku.dk (H.J.L.J.); 2New England Biolabs, Inc., 240 County Road, Ipswich, MA 01938, USA; sjohnson@neb.com; 3Department of Biochemistry and Molecular Biology, Michigan State University, 603 Wilson Rd, East Lansing, MI 48824, USA; hamberge@msu.edu

**Keywords:** endophyte, glycoalkaloids, phenolics, polyacetylenes, *Piriformospora indica*, secondary metabolism, terpenes, tomato

## Abstract

Specialised metabolites produced during plant-fungal associations often define how symbiosis between the plant and the fungus proceeds. They also play a role in the establishment of additional interactions between the symbionts and other organisms present in the niche. However, specialised metabolism and its products are sometimes overlooked when studying plant-microbe interactions. This limits our understanding of the specific symbiotic associations and potentially future perspectives of their application in agriculture. In this study, we used the interaction between the root endophyte *Serendipita indica* and tomato (*Solanum lycopersicum*) plants to explore how specialised metabolism of the host plant is regulated upon a mutualistic symbiotic association. To do so, tomato seedlings were inoculated with *S. indica* chlamydospores and subjected to RNAseq analysis. Gene expression of the main tomato specialised metabolism pathways was compared between roots and leaves of endophyte-colonised plants and tissues of endophyte-free plants. *S. indica* colonisation resulted in a strong transcriptional response in the leaves of colonised plants. Furthermore, the presence of the fungus in plant roots appears to induce expression of genes involved in the biosynthesis of lignin-derived compounds, polyacetylenes, and specific terpenes in both roots and leaves, whereas pathways producing glycoalkaloids and flavonoids were expressed in lower or basal levels.

## 1. Introduction

Fossils from the Early Devonian period displaying fungal-like structures inside plant cells provide evidence that plant-fungal interactions date back to more than 400 million years ago. In fact, it appears that these symbiotic interactions have contributed to the colonisation of land and the emergence of the first terrestrial plants [1,2,3]. Today, plant-fungal symbiosis is considered a dynamic interaction, ranging from mutualism and commensalism, to parasitism, depending on the result of the interaction on the plant host [4]. Mutualistic interactions in the soil are usually characterised by exchange of nutrients, during which the fungus acquires carbon from the host and, in return, it facilitates the acquisition and assimilation of essential and scarce soil elements (e.g., nitrogen, phosphorus, and iron) [3,5]. However, chemical crosstalk in plant-fungal symbioses is more complex and often serves multiple purposes. Plants produce defence molecules to fight pathogens or exude a variety of compounds to recruit beneficial fungi, while fungal symbionts synthesise metabolites to reprogram plant signalling and metabolism or antimicrobials to defend themselves from competing microorganisms [6,7,8]. Products of the specialised metabolism are often involved in these processes, shaping not only the actual plant-fungal symbiosis but also interactions with other organisms on a multitrophic level [9,10,11,12,13].

The root endophyte *Serendipita indica* (formerly known as *Piriformospora indica*, order Sebacinales, phylum Basidiomycota) has been extensively used in studying plant-fungal interactions [14,15,16]. *S. indica* is an axenically cultivable fungus, able to colonise a wide range of plant species. Colonisation by *S. indica* displays an unusual, biphasic pattern, according to which the fungus switches from the initial biotrophic growth to a necrotrophic-like behaviour, inducing programmed cell death in the roots [17,18]. During the early colonisation phase, *S. indica* secretes effector-like proteins and interferes with the plant hormone homeostasis to evade or suppress host defence [17,18,19]. Establishment of biotrophy at this stage is further assisted through the production of phytohormone analogues by the endophyte itself [20].

Colonisation by *S. indica* can promote plant growth and development [21,22] and enhance plant tolerance to biotic and abiotic stresses [23,24,25]. The exact mechanisms via which *S. indica* confers these benefits to the host are yet to be fully elucidated, but they appear to be related to the endophyte’s ability to improve nutrient uptake and induce plant defence responses [26,27,28]. For example, association with *S. indica* causes accumulation of several plant specialised metabolites [21,29,30,31,32], compounds that are often implicated in defence mechanisms [33].

In this study, we investigated a mutualistic plant-fungal symbiosis focusing on the dynamics of specialised metabolism in the host plant. Tomato plants (*Solanum lycopersicum*) were used as hosts due to their importance as a vegetable crop (worldwide production of 180 million tonnes [34]) and their well-characterised specialised metabolism [35,36,37]. In detail, roots and leaves of *S. indica*-colonised and endophyte-free tomato plants were subjected to a whole-transcriptome analysis using RNAseq. Our results showed that the presence of *S. indica* in the roots led to a strong transcriptional response in the leaves of colonised plants, which has not been reported before. In addition, we observed that *S. indica* colonisation caused accumulation of transcripts from genes involved in the biosynthesis of lignin-derived compounds, polyacetylenes, and specific terpenes in roots and leaves, while pathways producing glycoalkaloids and flavonoids were downregulated or left unaffected.

## 2. Materials and Methods

### 2.1. Fungal and Plant Material

*Serendipita indica* (isolate DSM11827) was cultured on solid complete medium (CM) plates supplemented with 1.5% (*w*/*v*) agar (*Aspergillus* medium, [38]) at 28 °C.

Tomato seeds (*Solanum lycopersicum*, cv. Moneymaker) were surface sterilised (1 min in 70% ethanol, 10 min in 1% NaClO (*v*/*v*)), rinsed multiple times with sterile MilliQ water, and then transferred in a growth chamber (12 h day 22 °C/12 h night 18 °C, 120 μE/m^2^·s light intensity, 60% relative humidity) until fully germinated (11 days). Tomato seedlings were incubated in a *S. indica* chlamydospore suspension (40 mL with a concentration of 300,000 chlamydospores/mL) overnight, on a shaker (120 rpm) at room temperature. For the control treatment, tomato seedlings were incubated in sterile water. Control and *S. indica*-inoculated seedlings were sown on Murashige–Skoog (MS) basal medium (Sigma-Aldrich, St. Louis, MO, USA), supplemented with 1.5% (*w*/*v*) agar and grown in the same growth chamber until harvest.

### 2.2. RNA Isolation from Tomato Leaves and Roots and RNA Sequencing

Tomato seedlings were harvested 11 dpi (days post inoculation) and roots and leaves (first and second true leaves) were sampled separately. Tissues from seven plants were pooled in one biological replication, and four biological replications were included for each treatment and each tissue. Before RNA extraction, the samples were freeze-dried overnight to facilitate milling. Total RNA was extracted using the Spectrum™ Plant Total RNA Kit (Sigma-Aldrich, St. Louis, MO, USA) according to the manufacturer’s instructions, and the samples were sent for sequencing to Novogene Bioinformatics Technology Co. Ltd. (Hong Kong). After sample RNA quality and integrity were verified on an Agilent 2100 Bioanalyser, 16 libraries (four biological replications per treatment per tissue) were constructed and processed by Illumina HiSeq sequencer. Sequencing was done on 150 bp paired-end reads, generating ≥30 million reads per sample (Appendix A). The raw reads were deposited in NCBI’s Sequence Read Archive (SRA) under the BioProject accession number PRJNA789108 (https://www.ncbi.nlm.nih.gov/bioproject/PRJNA789108).

### 2.3. Bioinformatic Analysis, Differential Expression, and Gene Ontology Enrichment Analysis

Raw reads were filtered to remove adapter sequences or reads of low quality using BBDuk [39]. Salmon version 0.12.0 [40] was used to estimate the abundance of *Solanum lycopersicum* transcripts (gene models from ITAG3.2), available in [41]. Differential expression analysis was performed with Deseq2 version 3.8 (false discovery rate, FDR < 0.1) [42]. Genes with an FDR-corrected *p* ≤ 0.05 were considered differentially expressed. The logarithmic fold change of gene expression between the treated and the control samples was used for comparative analysis of the data (Appendix A). Expression of genes involved in the main tomato metabolic pathways was presented in the form of a heatmap using R Studio.

Plant MetGenMAP package [43] was used to identify the enriched Gene Ontology (GO) terms related to biological processes that the differentially expressed genes belong to. Only the genes that were two-fold differentially expressed were used, and only the GO terms that were significantly enriched with a simulation-corrected *p* ≤ 0.05 were considered (Appendix A). Visualisation of selected GO enrichment analysis results was performed with R Studio [44] and scripts were generated by REVIGO [45].

Phytohormone biosynthesis, signalling, and specialised metabolism pathways of *S. lycopersicum* were assembled using a gene-by-gene approach, combining data from previously published literature and expression data from our study (Appendix A–S16).

## 3. Results

### 3.1. Summary of RNAseq Data and Differential Gene Expression Analysis

Of the reads from each sample, 80–87% were mapped to the tomato transcriptome (*Solanum lycopersicum* gene models ITAG3.2) (Appendix A). After comparing expression data between leaves of colonised and control plants, we found 6138 genes differentially expressed (FDR corrected *p* ≤ 0.05), among which the expression of 1259 genes was at least two-fold changed (391 genes upregulated and 868 downregulated). In the roots, only the transcript levels of 957 genes were affected, and almost one-third of them (300 genes) were expressed with a fold change equal or higher than two after colonisation (128 genes upregulated and 172 genes downregulated) (Figure 1).

### 3.2. GO Term Enrichment Analysis

Only the differential expressed genes in leaves and roots with at least two-fold change in expression were used for the GO term enrichment analysis. The most significantly over-represented GO terms among the upregulated genes in leaves were associated with RNA processing, ribosome biogenesis, and nuclear transport (GO:0042254 ribosome biogenesis, GO:0006364 rRNA processing, GO:0034470 ncRNA processing, GO:0006606 protein import into nucleus) (Appendix A). Regarding downregulated genes in leaves, the most significant enrichment in biological processes involved the wide category of transcription regulation (GO:0003700 DNA-binding transcription factor activity, GO:0030528 obsolete transcription regulator activity) (Appendix A).

Some of the most significantly enriched GO terms (simulation-corrected *p* ≤ 0.005) among the root upregulated genes belonged to processes related to defence (GO:0031347 regulation of defence response, GO:0010200 response to chitin) and plant immunity (GO:0045087 innate immune response, GO:0009627 systemic acquired resistance). Other enriched GO categories (simulation corrected *p* ≤ 0.05) were associated with hormone biosynthesis and signalling (GO:0009863 salicylic acid-mediated signalling pathway, GO:0042446 hormone biosynthetic process, GO:0009867 jasmonic acid-mediated signalling pathway), as well as programmed cell death (PCD) (GO:0010363 regulation of plant-type hypersensitive response, GO:0043067 regulation of programmed cell death) and nitrogen starvation processes (GO:0006995) (Appendix A). The most significantly over-represented GO terms among the downregulated genes in the roots were biological processes relevant to abiotic stress responses (GO:0009314 response to radiation, GO:0009416 response to light stimulus, GO:0009628 response to abiotic stimulus), as well as phenylpropanoid and flavonoid biosynthesis (GO:0009699 phenylpropanoid biosynthetic process, GO:0009813 flavonoid biosynthetic process) (Appendix A).

### 3.3. S. indica Root Colonisation Downregulates Glycoalkaloid Biosynthetic Genes in Tomato Leaves

The presence of *S. indica* in the roots of tomato plants resulted in downregulation of most of the genes involved in the synthesis of steroidal glycoalkaloids in leaves. The pathway leading to cholesterol, the steroidal glycoalkaloid precursor, and their core pathway (GAME: glycoalkaloid metabolism genes) were found to be less expressed in leaves of colonised plants compared to the leaves of control plants. However, only minor changes were observed in the roots, the place of the direct plant–endophyte interaction, where only *GAME12* and *GAME7* showed significant downregulation due to fungal colonisation (Figure 2, Appendix A).

### 3.4. Effect of S. indica Colonisation on Tomato Phenylpropanoid-Related Genes

*S. indica* colonisation barely modulated biosynthesis of phenolic compounds in both leaves and roots of tomato plants. The majority of the genes involved in the phenylpropanoid core pathway, as well as the genes involved specifically in the biosynthesis of flavonoids, flavonols, and anthocyanins, were either downregulated or not affected at all by *S. indica*. However, *C3H1* and *pCSE*, which direct the flux into the biosynthetic pathway of lignin and hydroxycinnamic acids (HCAs), were the only two genes significantly upregulated by the presence of the fungus in both tissues (Figure 3, Appendix A).

### 3.5. S. indica Upregulates Genes Involved in the Biosynthesis of Highly Modified Fatty Acids in Tomato

Among the 41 genes annotated as putative fatty-acid desaturases (FADs) in the tomato genome, seven genes in the roots and four in the leaves of colonised plants were significantly upregulated (Figure 4A). Although FADs are implicated in multiple physiological processes in plants (e.g., energy storage, membrane formation, hormone biosynthesis), the specific FADs found upregulated in tissues of colonised plants (Figure 4B) belong to a recently discovered gene cluster in tomato, responsible for the production of falcarindiol, a prototypical acetylenic lipid with antifungal properties [51].

### 3.6. S. indica Changes Expression of TPSs in Tomato Leaves and Roots

Root colonisation by *S. indica* only affected specific terpene synthase genes (*TPSs*) (Figure 5A, Appendix A) and transcription factors (Appendix A) potentially involved in terpenoid metabolism. Specifically, *TPS5* and *TPS9*, involved in linalool and germacrene C synthesis, respectively, were found to be downregulated in leaves of *S. indica*-colonised plants. On the contrary, we observed a high abundance of transcripts from *TPS51*, involved in the production of several sesquiterpenes (α-bisabolol, (*Z*)-nerolidol, α-bisabolene, β-bisabolene, and (*Z*)-β-farnesene) [52] in the leaves of colonised plants.

*S. indica* also induced transcript accumulation of three terpene synthase genes (*TPS20*, *TPS32*, *TPS33*) in tomato roots (Figure 5, Appendix A). *TPS20* encodes for an enzyme that produces mainly β-phellandrene and a mixture of other monoterpenes [53]. According to [54], *TPS32* catalyses the formation of viridiflorene and several unidentified sesquiterpenes, whilst *TPS33* forms several sesquiterpenes such as guaia-1(10),11-diene and β-acoradiene [52].

## 4. Discussion

### 4.1. Root Colonisation by S. indica Strongly Affected the Leaf Transcriptome

The effect of *S. indica* colonisation on the tomato transcriptome was studiedin tomato roots, the place of the direct interaction, and simultaneously in the leaves. To our knowledge, this is the first study where this approach was followed to investigate a microbial interaction with tomato plants. On the 11th day after inoculation with the endophyte, a total of 957 genes were found to be differentially expressed in the roots of tomato plants, while surprisingly approximately six times more genes (6138) were affected in the leaves. Endophytic root colonisation resulted in a major transcriptional response in the leaves, an unexpected observation since previous studies suggested that the endophyte barely affects aerial parts of colonised plants, in the absence of any other challenge [57,58]. Discrepancies between our results and results from earlier reports could be ascribed to differences between experimental setups and sampling points or the use of more sensitive techniques for studying transcriptional changes (RNAseq instead of microarrays). Unfortunately, there is a gap in studying simultaneously transcriptional responses of below- and aboveground parts of plants associated with microorganisms. More studies investigating the effect of endophytic colonisation not only at the site of the direct interaction, but also at distant host tissues are necessary to unravel its complexity and potential.

### 4.2. GO Term Enrichment Analysis in Roots Indicates S. indica Lifestyle and Colonisation Phase

The GO term analysis of the leaf data did not give any obvious information about the biological processes affected in the aerial parts of the plant by *S. indica* colonisation. It is possible, however, that enrichment of upregulated genes in categories such as RNA processing, ribosome biogenesis, and nuclear transport, usually enhanced during cell proliferation and plant development [59], could indicate shoot growth promotion due to the presence of *S. indica* as reported in previous studies [21,23]. On the contrary, GO term analysis in roots showed that, in addition to defence and biotic stress responses, categories such as programmed cell death, hypersensitive response (HR), and respiratory burst responses against pathogen attacks were overrepresented in root upregulated genes. It has been previously shown that root colonisation by *S. indica* displays an unusual, biphasic pattern, during which the fungus biotrophically colonises the root in the early stages of infection (1–2 dpi), while, in later stages, a switch to a necrotrophic-like behaviour is observed, with the fungus proliferating and sporulation in dead root cells [17,18]. In accordance with this, the GO term analysis results indicated that, at the specific sampling point in the experiment (11 dpi), *S. indica* has already entered its necrotrophic phase of colonisation, upregulating genes involved in programmed cell death responses.

### 4.3. S. indica Negatively Regulates Glycoalkaloid Biosynthesis in Leaves of Colonised Plants

Steroidal alkaloids and steroidal glycoalkaloids are nitrogen-containing compounds, produced by solanaceous plants and involved in plant defence against pathogens and pests [60,61]. The most abundant steroidal alkaloids in tomato green tissues are α-tomatine and its precursor dehydrotomatine [62] and the main enzymes and genes involved in their biosynthesis have been identified [46,47,48].

Our gene-by-gene approach of studying thetomato specialised metabolism showed that *S. indica* colonisation resulted in a decreased expression of the genes of cholesterol biosynthesis and the GAME (glycoalkaloid metabolism) genes involved in glycoalkaloid production in the leaves of tomato plants. One of the genes downregulated was *GAME9* (Appendix A), an APetala2/ethylene-responsive factor (AP2/ERF) transcription factor that regulates the biosynthesis of these compounds in Solanaceae plants [47]. In *Catharanthus roseus*, biosynthesis of terpenoid indole alkaloids is also regulated by an AP2/ERF MeJA-responsive transcription factor, ORCA3. CrMYC2 is the major activator of *ORCA3* expression, and knocking it down results in a reduction in *ORCA3* transcript levels [63]. A similar mechanism linking *SlMYC2, GAME9*, and alkaloids is also proposed for tomato [47]. *S. indica* causes *SlMYC2* (Solyc08g076930.1.1) downregulation (Appendix A, fold change = −0.71, FDR-corrected *p* ≤ 0.05) in the leaves of colonised plants, due to a general suppression observed in the JA pathway. This possibly also causes *GAME9* downregulation and, thus, the decreased expression of the full SGA biosynthetic machinery. Regarding the roots, the expression of only two glycoalkaloid-related genes was negatively affected. Low levels of α-tomatine and its precursor in colonised roots could allow for the expansion of the fungus since both compounds act against many fungi [62].

In a broader view, regulation of glycoalkaloid production has been connected with nitrogen availability in many plants [64]. According to our analysis, genes involved in nitrogen starvation responses are among the most enriched categories in the root upregulated genes (Appendix A-GO:0006995, cellular response to nitrogen starvation), indicating that *S. indica* colonisation causes nitrogen depletion in the roots of tomato plants. This decrease in nitrogen levels, also observed in *S. indica*-colonised barley plants [65], could have generated signals, sent throughout the plant, in order to save up host nitrogen resources by suppressing related pathways, including glycoalkaloid biosynthesis.

### 4.4. S. indica Colonisation Causes Downregulation of Most Phenylpropanoid Biosynthetic Genes in Leaves of Tomato Plants

Phenylpropanoid compounds comprise a diverse group of plant specialised metabolites, also found in tomato. Phenylpropanoids include several groups of compounds such as flavonoids and flavonols, lignans, hydroxycinnamic acids, and their derivatives. In tomato, these compounds play various roles, serving in abiotic stress tolerance (flavonols) [49] or acting as antioxidants (chlorogenic acid) [66] and structural components (lignin and suberin) [67].

The GO term enrichment analysis showed that processes related to phenylpropanoid and flavonoid biosynthesis were two of the most over-represented categories among the root downregulated genes (Appendix A). Following a more detailed study of some of the individual genes involved, we sawthat the fungus causes slight downregulation of the phenylpropanoid and related pathways, e.g., flavonoids, flavonols, and anthocyanins, mainly in the leaves of colonised plants (Figure 3, Appendix A). The only two genes upregulated, in both roots and leaves, were *C3H* and *CSE* (Figure 3, Appendix A). These genes encode enzymes that use *p*-coumaric acid and *p*-coumaroyl-CoA, respectively, to synthesise hydroxycinnamic acids, which could be further used in lignin biosynthesis.

Our results do not align with other studies showing that endophytic colonisation by *S. indica* can induce production of some phenylpropanoids [21,58,68]. According to a nontargeted analysis done on several tissues of *Arabidopsis* plants inoculated with *S. indica* (14 dpi), the phenylpropanoid pathway was among the most enriched transcriptionally up-regulated KEGG pathways and several compounds (coumarins, oligolignols, and flavonoids) accumulated in the metabolite profiles of colonised roots. However, only oligolignols and hydroxycinnamic acid amides, and not flavonoids, were overproduced in leaves of colonised plants. Due to this observation, it was concluded that flavonoids are important key players in the *A. thaliana* and *S. indica* mutualistic interaction [58], which may not apply for the association of *S. indica* with tomato, a plant with a very different specialised metabolism. In addition, our data represent only a snapshot of the regulation of plant metabolism, and more sampling points may be necessary to shed light to whether these changes in gene expression are species-specific or time-related. In accordance with our findings on the increased accumulation of transcripts related to lignin and HCAs biosynthesis, lignan podophyllotoxin and several phenolic acids were found accumulated in hairy roots of *Linum album* that were treated with *S. indica* mycelium extract [69]. Increased flux towards the production of these phenolic compounds appears to be more consistent across the different host species [32,58,69], indicating a generalised response to *S. indica* colonisation. This response could be further associated with defence mechanisms or enhanced water use efficiency and drought tolerance, through xylem strengthening, previously observed in *S. indica*-colonised plants [21].

### 4.5. Genes Involved in Tomato Polyacetylene Biosynthesis Are Induced by S. indica

A group of transcripts annotated as fatty acid desaturases (FADs) were among the most upregulated genes in colonised tomato roots (Figure 4A, Appendix A). Initially, these genes were considered to be involved in oxylipin and jasmonate biosynthesis, a response generally induced by *S. indica* colonisation and previously observed [19,68]. However, since our data suggested that JA biosynthesis and signalling were not affected at transcriptional level by the colonisation (Appendix A), we investigated the role of these genes in other pathways.

In [70], the biosynthetic steps of carrot polyacetylenes, lipid compounds produced after pathogen attack, were elucidated. Although polyacetylenes are produced mainly in the Apiaceae and Araliaceae families, the polyacetylenic products falcarindiol and falcarinol have been also detected in the leaves of domesticated tomatoes after infection by the fungus *Cladosporium fulvum*. Due to their antifungal activities, these compounds are considered to act in plant defence [35]. The biosynthetic pathway of acetylenic fatty acids shares the same starting point as linolenate and oxylipin biosynthesis. However, instead of the oleate conversion to linolenate by a Δ12 fatty acid desaturase (FAD), “divergent” FADs add unusual functional groups into the acyl chain (epoxyl or hydroxyl groups, conjugated double bonds, triple bonds), resulting in these highly modified fatty acids [71]. According to the recently elucidated biosynthetic steps of tomato polyacetylenes [51], the FADs found upregulated by *S. indica* colonisation belong to this group of these “divergent” enzymes, involved in the production of specialised metabolites (e.g., falcarinol and falcarindiolin) (Figure 4B). This could be an indication that the plant induces production of antifungal compounds in the roots in order to stop extensive growth of *S. indica*. In addition, this study shows that expression of this biosynthetic gene cluster can also be activated by a non-pathogenic fungus, which has not been reported previously.

### 4.6. Expression of a Recently Described TPS Is Induced in the Leaves of S. indica-Colonised Plants

One additional group of genes affected in fungal colonised plants belonged to the terpene biosynthetic pathway. Terpenes are classified as mono- (C10), sesqui- (C15), di- (C20), tri-terpenes (C30), etc., depending on the number of carbon atoms they are composed of. Terpenes can contribute to plant direct [72] or indirect defences [73]. Biosynthesis of tomato terpenes has been extensively studied in tomato with key enzymes and transcription factors of the pathway widely identified [52,53,54,55,56,74,75,76,77].

The presence of *S. indica* was found to affect only specific *TPS*s in roots and leaves (Figure 5A, Appendix A). In colonised roots, the upregulated *TPS20* encodes a β-phellandrene synthase that produces, together with β-phellandrene, a mixture of other monoterpenes [53]. Two more genes, localised in close proximity on chromosome 1, were expressed in the colonised roots, i.e., *TPS32* and *TPS33*. The specific genes were also found to be upregulated in tomato fruits from plants colonised by arbuscular mycorrhizal fungi [78]. In addition to the biosynthesis of several sesquiterpenes that they individually catalyse, *TPS32* and *TPS33* (together with *TPS31*) comprise a biosynthetic cluster considered to be involved in the synthesis of rishitin, a tomato defence compound [78]. This hypothesis is further supported by the fact that the amino-acid sequences of these genes are closely related to the pepper 5-*epi*-aristolochenesynthase (EAS) involved in capsidiol biosynthesis, the main defence metabolite produced in peppers, equivalent to rishitin. In addition, we found a gene overexpressed in colonised roots (Solyc11g007980.2.1), which is the closest orthologue of the *Capsicum annuum* 5-*epi*-aristolochene-1,3-dihydroxylase gene (EAH) in tomato, and also involved in the same pathway [79].

Another study implicated high expression of *TPS31*, *TPS32*, *TPS33*, and *TPS35* in leaves with increased resistance to *Ralstonia solanacearum* in the tomato cultivar H7996. In the same study, silencing of these genes in tomato plants resulted in an increased proliferation of *R. solanacearum* compared to control plants also infected with the bacterium, but also to a decrease in expression of SA and ET signalling markers [80]. This suggests that these genes and their products play a crucial role in plant defence.

*S. indica* presence has coincided previously with increased expression of terpene synthases [81]. In barley roots, *S. indica* colonisation appeared to induce expression of kaurene synthases, involved in biosynthesis of phytohormones and defence specialised metabolites. In addition, silencing of these genes in mutant lines resulted in a reduced colonisation by the endophyte, suggesting that the compounds produced play a role in establishing a successful interaction between the plant and the endophyte [81].

In the leaves of colonised plants, two terpene synthase genes were downregulated (*TPS5* and *TPS9*), probably due to downregulation of a transcription factor that regulates their expression (*SlMYC1*, Solyc08g005050.3.1) [82] (Appendix A). Additionally, a recently characterised sesquiterpene synthase gene, *TPS51*, was the only one significantly upregulated in leaves of *S. indica*-inoculated plants. Transcripts of this gene have been detected previously in leaf tissues treated with various pathogens [83]. *TPS51* encodes a terpene synthase that catalyses the formation of mostly (*E*)-nerolidol from (*E,E*)-FPP and several sesquiterpenes (α-bisabolol, (*Z*)-nerolidol, α-bisabolene, β-bisabolene, and (*Z*)-β-farnesene) from (*Z,Z*)-FPP [52]. Although there is no ecological role assigned to these compounds, it is likely that they could act as infection alarms, warning neighbouring plants about microbial intruders in the roots [84,85].

## 5. Conclusions

Plant specialised metabolites are important for adaptation and interaction with the environment, serving as defence compounds against microorganisms or other competing plants, or as signal molecules, attracting pollinators or seed dispersal animals. However, this class of compounds is activated not only in case of pathogen or predator attacks but also after endophytic colonisation, as seen here. In this case, tomato plants respond to *S. indica* colonisation by overexpressing specific biosynthetic pathways of lignin-derived compounds, polyacetylenes, and terpenes and by lowering expression of glycoalkaloid and phenylpropanoid biosynthesis. This transcriptional regulation of plant specialised metabolism in the roots can generally be explained as an attempt by the host to slow down uncontrolled expansion of the fungus by activating production of certain defence compounds and to save up resources by shutting down others. The general signalling alarm starting in the roots due to *S. indica* colonisation and transferring the information of a root intruder throughout the whole plant could be the reason why we observed similar expression patterns in the leaves of colonised plants. This activation or downregulation of specialised metabolism in the aerial parts of an endophyte-colonised plant can affect symbiotic relations with organisms of the surrounding environment but it can also be used in developing diagnostic tools that detect microbial infection/colonisation. Since establishment of any plant–microbe association involves communication through chemical compounds, it is crucial to understand how their biosynthesis is regulated and elucidate their role in the interaction in order to fully understand symbiotic events.

## Figures and Tables

**Figure 1 microorganisms-10-00194-f001:**
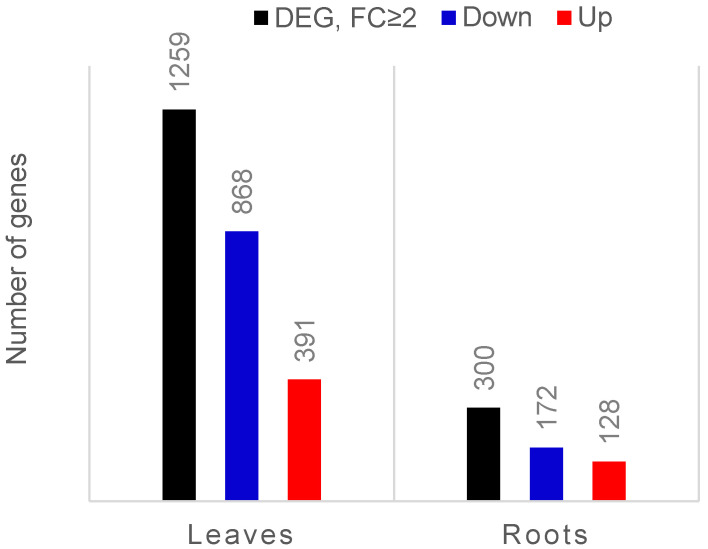
Bar chart showing the number of differentially expressed genes (DEGs; FDR-corrected *p*-value ≤ 0.05) in leaves and roots of endophyte-colonised compared to endophyte-free tomato plants. The black bars represent the total number of DEGs that exhibited at least two-fold changed expression (FC ≥ 2). Upregulated genes are marked in red and downregulated genes are marked in blue.

**Figure 2 microorganisms-10-00194-f002:**
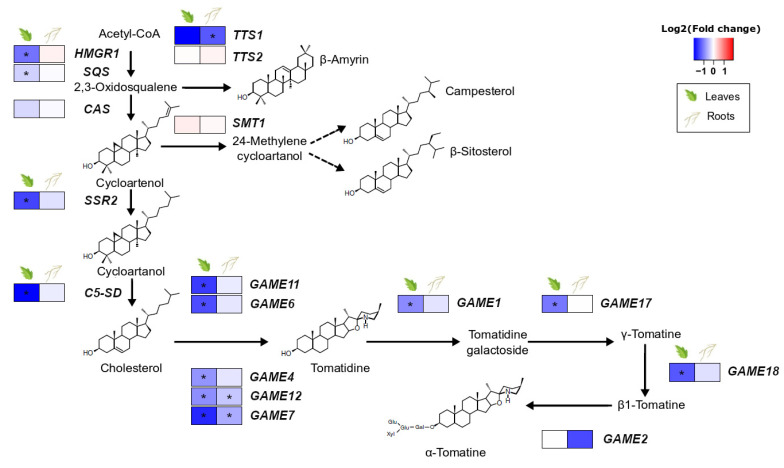
Effect of *S. indica* colonisation on the expression of genes involved in the biosynthetic pathway of sterol precursors and glycoalkaloids in tomato leaves and roots. Changes in gene expression between tissues from inoculated plants and tissues from endophyte-free plants are represented as log_2_ of the fold change in the form of a heatmap. Data derived from RNA sequencing of leaves and roots from *S. indica*-colonised and endophyte-free plants. Asterisks (*) indicate FDR-corrected *p* ≤ 0.05. HMGR1, 3-hydroxy-3-methylglutaryl-CoA reductase; SQS, squalene synthase; CAS, cycloartenol synthase; SSR2, sterol side-chain reductase 2; C5/SD, Δ(7)-sterol-C5-desaturase; TTS1, triterpenoid synthase 1; TTS2, triterpenoid synthase 2; SMT, sterol methyltransferase; adaptedfrom [46,47,48].

**Figure 3 microorganisms-10-00194-f003:**
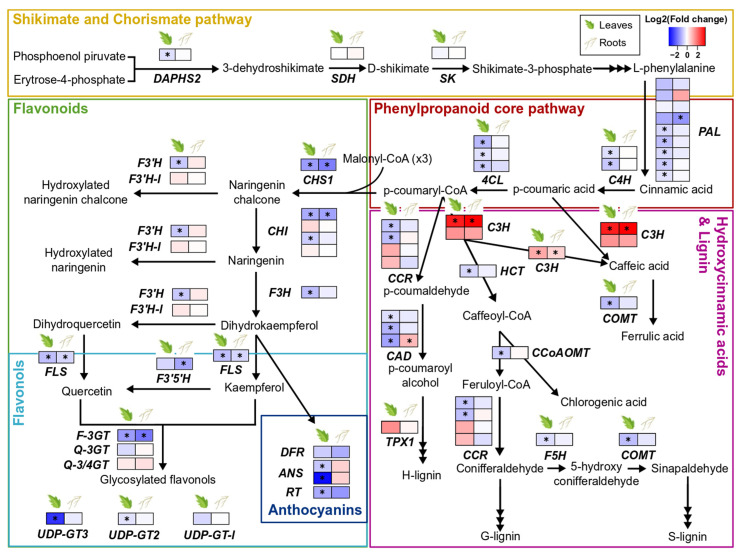
Changes in gene expression of phenylpropanoid biosynthetic pathways due to *S. indica*-colonisation in tomato leaves and roots. Key genes involved in the biosynthesis of shikimate and chorismate, flavonoids, flavonols, anthocyanins, hydroxycinnamic acids, and lignin are shown here. Changes in gene expression between tissues from inoculated plants and tissues from endophyte-free plants are represented as log_2_ of the fold change in the form of a heatmap. Asterisks (*) indicate FDR-corrected *p* ≤ 0.05. DAHPS2, DAHP synthase 2; SDH, shikimate 5-dehydrogenase; SK, shikimate kinase; PAL, phenylalanine ammonia lyase; C4H, cinnamate 4-hydroxylase; 4CL, 4-coumarate:CoA ligase; CHS1, chalcone synthase 1; F3′H, flavonoid-3′-hydroxylase; F3′H-l, flavonoid-3′-hydroxylase-like; CHI, chalcone isomerase; F3H, flavanone-3-hydroxylase; FLS, flavonol synthase; F3′5′H, flavonoid-3′ 5′-hydroxylase; F-3GT, flavonol 3-glycosyltransferase; Q-3GT, quercetin 3-glycosyltransferase; Q-3/4GT, quercetin 3/4-glycosyltransferase; UDP-GT3, UDP-glycosyltransferase 3; UDP-GT2, UDP-glycosyltransferase 2; UDP-GT-l, UDP-glycosyltransferase-like; DFR, dihydroflavonol reductase; ANS, anthocyanidin synthase; RT, anthocyanin 3-glycosyltransferase; CCR, cinnamoyl-CoA reductase; HCT, hydroxy-cinnamoyl transferase; CAD, cinnamyl alcohol dehydrogenase; TPX1, tomato peroxidase 1; C3H, *p*-coumaroyl 3-hydroxylase; pCSE, putative caffeoyl shikimate esterase; COMT, caffeic acid *O*-methyltransferase; CoAOMT, caffeoyl CoA 3-*O*-methyltransferase; F5H, coniferyl aldehyde 5-hydroxylase; adapted from [49,50].

**Figure 4 microorganisms-10-00194-f004:**
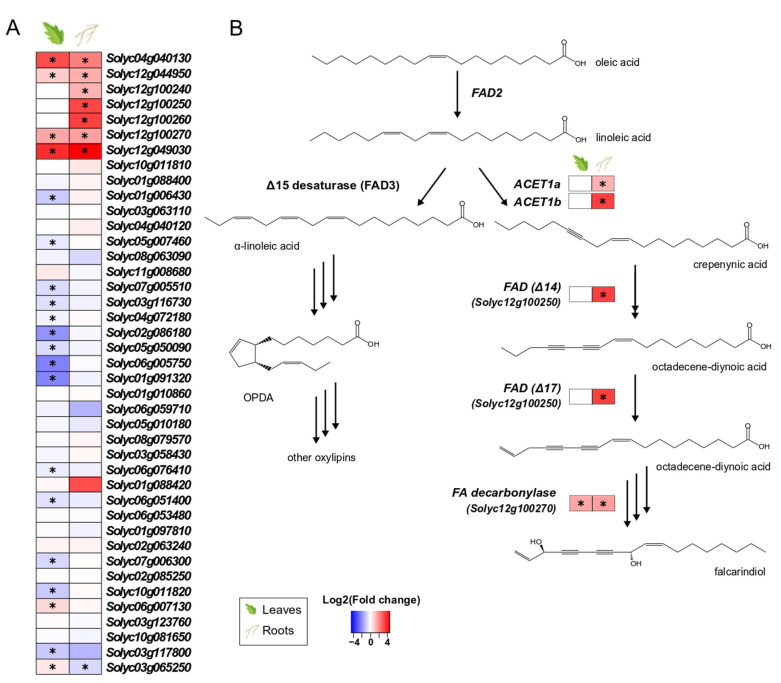
Tomato putative fatty-acid desaturases (FADs) involved in the biosynthesis of fatty acids in leaves and roots. (**A**). Changes in gene expression between tissues from inoculated plants and tissues from endophyte-free plants are represented as log_2_ of the fold change in the form of a heatmap. Data derived from RNA sequencing on leaves and roots from *S. indica*- and mock-inoculated plants. Asterisks (*) indicate FDR-corrected *p* ≤ 0.05. (**B**). Biosynthetic pathway leading to oxylipins (left) and falcarindiol (right). Linoleic acid is used in the biosynthesis of jasmonic acid and other oxylipins, as well as the production of highly modified fatty acids (polyacetylenes). Proposed chemical structures, gene function and biosynthetic pathway are based on the work of [51].

**Figure 5 microorganisms-10-00194-f005:**
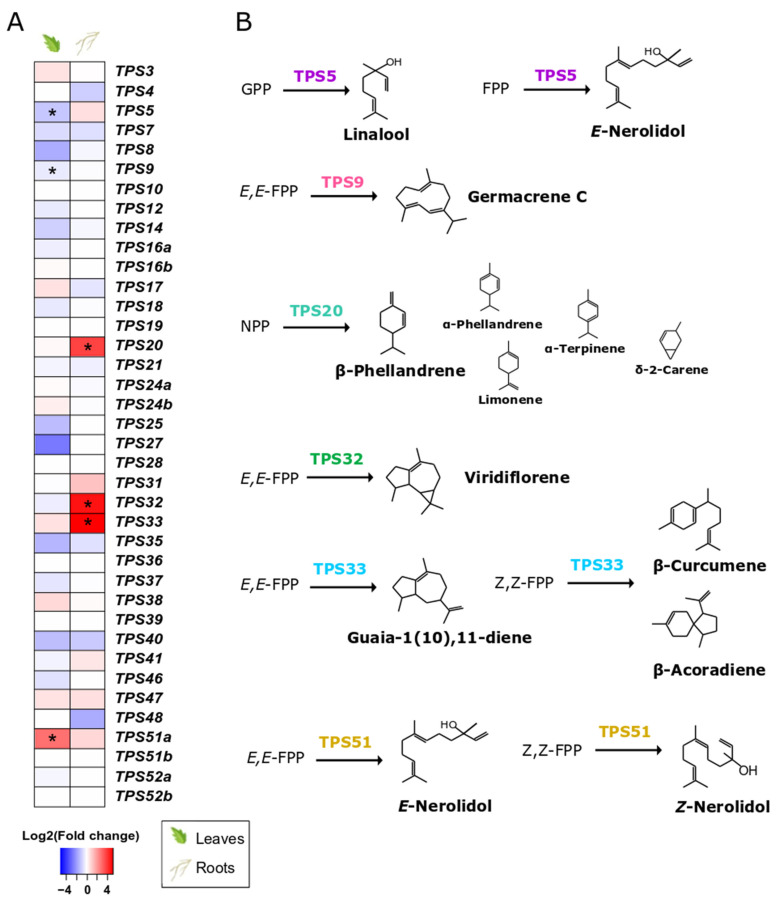
The tomato TPS family. (**A**). Effect of *S. indica* colonisation on the expression of tomato *TPS*s in leaves and roots. Changes in gene expression between tissues from inoculated plants and tissues from endophyte-free plants are represented as log2 of the fold change in the form of a heatmap. Data derived from RNA sequencing on leaves and roots from *S. indica*- and mock-inoculated plants. Asterisks (*) indicate FDR-corrected *p* ≤ 0.05, apart from *TPS32* and *TPS33*, whichwere expressed only in samples from inoculated roots, and no related transcripts were found in the samples of control roots. (**B**). Biosynthetic pathways catalysed by the TPSs, the expression of which is affected by *S. indica* colonisation, are only presented here. Main products of the enzymatic reactions are shown in regular font size, while terpenoids detected as minor peaks are shown in a smaller size. Reactions are drawn based on data from *in vitro* enzymatic assays from [52,53,54,55,56].

## Data Availability

Raw data in the form of FASTQ files generated by the RNA sequencing are deposited in the NCBI Sequence Read Archive (SRA) under the BioProject accession number PRJNA789108 (https://www.ncbi.nlm.nih.gov/bioproject/PRJNA789108).

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
