# Peer review of "Regulation of Tomato Specialised Metabolism after Establishment of Symbiosis with the Endophytic Fungus Serendipita indica"

_microorganisms, 2022, doi:10.3390/microorganisms10010194_

Round 1
Reviewer 1 Report
All comments are in the attached file

Reviewer 2 Report
Ntana et al. did a transcriptome analysis of tomato seedlings inoculated or not with the root endophytic fungus Serendipita indica. The authors compared leaves and roots with and without fungal inoculation and describe the key pathways modulated after inoculation. The study was well-done and the description of results is reasonable. However, the scientific soundness of the manuscript would be enhanced by validating at least some of the fungus-induced changes in gene expression in an independent experiment. Furthermore, the significance of the results would increase if the observed change in gene expression of the phenylpropanoid pathway and the sterol and glykoalkaloid pathway could be correlated with changes in levels of specific metabolites. This would be particularly interesting for the production of tomatine, but also for some of the fatty acids that are assumed to be differentially enriched based on the enhanced expression of putative fatty acid desaturases.
The results are overall well discussed, though I miss a hypothesis why in the present study - in contrast to previous studies - changes in transcript levels were mainly observed in the shoot and not the root (line 258-259). Similarly, I miss an explanation/hypothesis why your results are different from S. indica colonized Arabidopsis plants showing an enhanced induction of the phenylpropanoid metabolism.
Minor comment:
It would be great if the references could be unified - currently some titles are written in capital letters, others not.
